# Single Cell Analysis of Bistable Expression of Pathogenicity Island 1 and the Flagellar Regulon in *Salmonella enterica*

**DOI:** 10.3390/microorganisms9020210

**Published:** 2021-01-20

**Authors:** María Antonia Sánchez-Romero, Josep Casadesús

**Affiliations:** 1Departamento de Microbiología y Parasitología, Facultad de Farmacia, Universidad de Sevilla, Calle Profesor García González, 2, 41012 Seville, Spain; 2Departamento de Genética, Facultad de Biología, Universidad de Sevilla, Apartado 1095, 41080 Seville, Spain; casadesus@us.es

**Keywords:** phenotypic heterogeneity, bistability, epithelial cell invasion, SPI-1, flagellar regulon

## Abstract

Bistable expression of the *Salmonella enterica* pathogenicity island 1 (SPI-1) and the flagellar network (Flag) has been described previously. In this study, simultaneous monitoring of OFF and ON states in SPI-1 and in the flagellar regulon reveals independent switching, with concomitant formation of four subpopulations: SPI-1^OFF^ Flag^OFF^, SPI-1^OFF^ Flag^ON^, SPI-1^ON^ Flag^OFF^, and SPI-1^ON^ Flag^ON^. Invasion assays upon cell sorting show that none of the four subpopulations is highly invasive, thus raising the possibility that Flag^OFF^ cells might contribute to optimal invasion as previously proposed for SPI-1^OFF^ cells. Time lapse microscopy observation indicates that expression of the flagellar regulon contributes to the growth impairment previously described in SPI-1^ON^ cells. As a consequence, growth resumption in SPI-1^ON^ Flag^ON^ cells requires switching to both SPI-1^OFF^ and Flag^OFF^ states.

## 1. Introduction

Invasion of epithelial cells by *Salmonella enterica* requires the activity of effectors secreted through the type III secretion system (T3SS) of pathogenicity island 1 (SPI-1). In the mammalian ileum and under reductionist laboratory conditions that mimic the intestine, expression of SPI-1 genes is bistable, yielding SPI-1^OFF^ and SPI-1^ON^ subpopulations [1,2,3]. The mechanism that controls SPI-1 bistability remains unknown but probably depends on complex interactive networks and feedback loops involving SPI-1 regulators [4]. Some such regulators control crosstalk between SPI-1 and other gene networks, and an example is the flagellar regulon [4,5,6].

A peculiar trait of SPI-1^ON^ cells is reduced growth rate [7], caused by the energetic burden of building the secretion apparatus, by disruption of the proton gradient during T3SS assembly, or by the energy expense invested in both processes. However, growth retardation associated to SPI-1 expression confers increased antibiotic resistance to SPI-1^ON^ cells [8].

Bacterial motility contributes to invasion by fostering bacterial contact with host cells [9] and by helping to choose optimal infection sites [10,11]. Flagella-mediated adhesion may also contribute to triggering membrane ruffling and concomitant initiation of epithelial cell invasion [12]. Furthermore, flagellin has been shown to be a potent activator of the innate immune response [13,14]. Like SPI-1, the *S. enterica* flagellar regulon shows bimodal expression [15], and formation of flagellated and non-flagellated subpopulations is under the control of multiple regulators [11,15,16,17,18]. 

While the contribution of SPI-1 and the flagellar network to epithelial cell invasion has been studied with exquisite detail, certain questions have remained unanswered because they demanded single cell analysis. One such question is whether SPI-1 and the flagellar regulon undergo independent or co-ordinated switching; another is whether flagellar expression, alone or together with SPI-1 expression, alters *S. enterica* growth. In this study, simultaneous monitoring of OFF and ON states in SPI-1 and the flagellar regulon has permitted visualization of four subpopulations (SPI-1^OFF^ Flag^OFF^, SPI-1^OFF^ Flag^ON^, SPI-1^ON^ Flag^OFF^, and SPI-1^ON^ Flag^ON^), indicating that independent switching occurs. In vitro assays using sorted (sub)populations have provided evidence that cells that express SPI-1 and flagella are not optimally invasive, suggesting that both Flag^OFF^ and Flag^ON^ cells may contribute to optimal epithelial cell invasion. We also show that expression of the flagellar regulon contributes to the growth impairment of SPI-1^ON^ cells.

## 2. Materials and Methods

### 2.1. Bacterial Strains, Media and Culture Conditions

Strains of *Salmonella enterica* serovar Typhimurium used in this study derive from the mouse-virulent strain SL1344. For simplicity, *Salmonella enterica* serovar Typhimurium is abbreviated as *S.* Typhimurium throughout the text. Strain SV7884 (*sipB*::*gfp*) was described elsewhere [3]. A transcriptional fusion with the *mCherry* gene was constructed downstream of the stop codon of *fliC*. For this purpose, a DNA fragment containing the promoterless *mCherry* gene and the kanamycin resistance cassette was PCR-amplified from pDOC-R, an *mCherry*-containing derivative of plasmid pDOC (a gift from Busby’s lab, Univ. Birmingham, UK), using primers 5’GAA CCA GGT TCC GCA AAA CGT CCT CTC TTT ACT GCG TTA AGG GTA CCA TGG TGA GCA AGG 3’ and 5’CTG CCT TGA TTG TGT ACC ACG TGT CGG TGA ATC AAT CGC CAA CCT CGA GAT ATG AAT ATC 3’. The PCR product was integrated into the chromosome of SL1344 using the Lambda Red recombination system [19], generating strain SV7819 (*fliC*::*mCherry*). P22 HT-mediated transduction was used to generate strain SV8533 (*sipB*::*gfp fliC*::*mCherry*) using SV7819 (*fliC*::*mCherry*) as donor and SV7884 (*sipB*::*gfp*) as recipient. Cultures were grown at 37 °C in borosilicate tubes containing 5 mL of Bertani’s lysogeny broth (LB). Oxygen limitation (“invasive” condition) was achieved by growth without shaking. 

### 2.2. Invasion Assays in HeLa Epithelial Cells

HeLa human epithelial cells (ATCC CCL-2) were grown in DMEM containing 10% fetal calf serum and 1 mM glutamine (Life Technologies, Carlsbad, CA, USA). HeLa cells were seeded in 24-well plates (Costar, Corning) the day before the infection. Bacterial cultures were grown overnight at 37 °C in LB without shaking in borosilicate tubes and added to HeLa cells to reach a multiplicity of infection (MOI) of 75 bacteria per eukaryotic cell. Thirty minutes after infection, cells were washed twice with phosphate buffered saline (PBS) and incubated in fresh DMEM medium containing 100 μg/mL gentamicin for 90 min. Numbers of viable intracellular bacteria were obtained after lysis of infected cells with 1% Triton X-100, and plating on appropriate media. Infections were carried out in triplicate. Invasion rates were calculated as the ratio between viable intracellular bacteria and viable bacteria added to infect the HeLa cells.

### 2.3. Flow Cytometry Analysis

Flow cytometry was used to monitor expression of *sipB*::*gfp* (SPI-1) and *fliC*::*mCherry* (flagella) fusions. Data acquisition was performed using a MACSQuant^®^ Analyzer 10 flow cytometer (Miltenyi Biotec, Bergisch Gladbach, Germany) and was analyzed with FlowJo X v. 10.0.7r software (Tree Star, Inc., Ashland, OR, USA). *S.* Typhimurium strains were grown at 37 °C until the desired optical density, washed, and re-suspended in PBS for fluorescence measurement by flow cytometry. Fluorescence values for 100,000 events were compared with the data from the reporter-less control strain, thus yielding the fraction of ON and OFF cells in SPI-1 and in the flagellar system. 

### 2.4. Fluorescence Activated Cell Sorting (FACS) of Live Cells

Cells from an overnight culture were grown under invasive conditions. The culture was washed and re-suspended in PBS to a final concentration of 5 × 10^6^ cells/mL. Cells were sorted using a MoFlo Astrios EQ cytometer (Beckman Coulter, Brea, CA). Immediately prior to sorting, cells were analyzed for GFP and/or mCherry expression. Based on this analysis, gates were drawn to separate cells that expressed both GFP (SPI-1) and mCherry (flagella) (SPI-1^ON^ Flag^ON^ subpopulation) from cells that did not (SPI-1^OFF^ Flag^OFF^ subpopulation), as well as cells that expressed either GFP (SPI-1) or mCherry (flagella) (SPI-1^ON^ Flag^OFF^ and SPI-1^OFF^ Flag^ON^ subpopulations). From each gate, cells were collected into a sterile tube. After sorting, cells were spun at 4000 rpm for 10 min. FACS buffer was then removed, and cells were re-suspended in DMEM to perform invasion assays. An aliquot of sorted cells was run again at the cytometer to confirm the purity of the preparation. Data were analyzed with FlowJo X v. 10.0.7r software (Tree Star, Inc.).

### 2.5. Fluorescence Microscopy

Strain SV8533 (*sipB*::*gfp fliC*::*mCherry*) was grown in LB without shaking for 18 h at 37 °C. Samples of 1.5 mL were collected by centrifugation at 3400× *g* for 5 min. Cells were placed on an agarose slab (0.9% agarose/1% LB medium) pre-warmed at 37 °C. Images were captured with a Zeiss Apotome fluorescence microscope equipped with a 100× Plan Apochromat objective lens and an incubation system that permits cultivation and observation of living cells (37 °C). Pictures were taken at different times using an Axiocam 506 camera, and the images were analyzed using ImageJ software (Wayne Rasband, Research Services Branch, National Institute of Mental Health, MD, USA).

### 2.6. Statistical Analysis

Student’s *t*-test (two-tailed) was performed using GraphPad Prism v. 6.0 for Mac to determine statistical differences between two groups.

## 3. Results

### 3.1. Single Cell Analysis of the Expression Pattern of Invasion and Motility Genes

Simultaneous monitoring of OFF and ON states in SPI-1 and the flagellar regulon was performed using transcriptional fusions with GFP and mCherry fluorescent proteins. The fusions had been constructed downstream of the SPI-1 gene *sipB*, which encodes a T3SS effector, and downstream of the flagellar gene *fliC,* which encodes flagellin. The resulting strain (SV8533, *sipB*::*gfp fliC*::*mCherry*) was used to monitor the pattern of SPI-1 and flagellar expression using fluorescence microscopy and flow cytometry in cultures grown in invasive conditions (LB under oxygen limitation). Bistable expression of both SPI-1 and the flagellar system was observed (Figure 1 and Appendix A). The subpopulation of SPI-1^ON^ cells was small as previously described [3] while the Flag^ON^ subpopulation was large. Invasion and motility genes thus appear showing independent patterns of bimodal expression, and the subpopulation of cells that secrete SPI-1 invasion effectors and harbor flagella is small. Bistability was also observed under non invasive conditions (LB under aerobiosis) but the subpopulation sizes were different (Appendix A).

### 3.2. Contribution of Virulence and Motility Systems to Invasion of Epithelial Cells In Vitro

Given the involvement of flagella in early steps of the invasion process, one might expect that Flag^ON^ cells would be more invasive than Flag^OFF^ cells [15]. To test this hypothesis, epithelial cell invasion assays were performed using Flag^OFF^ and Flag^ON^ subpopulations separated by cell sorting. Surprisingly, the Flag^OFF^ subpopulation was as invasive as the Flag^ON^ subpopulation (Figure 2B). Therefore, at first sight, cells that harbor flagella do not seem to be more invasive than cells that lack flagella. However, both Flag^OFF^ and Flag^ON^ subpopulations contain SPI-1-expressing and SPI-1-nonexpressing cells, and both are necessary for optimal invasion [3]. When sorted SPI-1^OFF^ Flag^OFF^, SPI-1^ON^ Flag^OFF^, SPI-1^OFF^ Flag^ON^ and SPI-1^ON^ Flag^ON^ subpopulations were tested in epithelial cell invasion assays, none of the subpopulations was invasive on its own (Figure 2C). However, the sorted SPI-1^ON^ subpopulations (SPI-1^ON^ Flag^OFF^ and SPI-1^ON^ Flag^ON^) showed higher invasion rates than the SPI-1^OFF^ subpopulations (SPI-1^OFF^ Flag^OFF^ and SPI-1^OFF^ Flag^ON^) (Figure 2C). The possibility that cell sorting might damage either the T3SS or the flagellum has been ruled out previously [3,20]. Furthermore, as a cautionary measure, the population labeled ALL (the whole bacterial population including the four subpopulations) was passed through the cell sorter before the invasion assays. The unsuspected observation that cells that express SPI-1 and flagella are not optimally invasive might perhaps indicate that both Flag^OFF^ and Flag^ON^ cells are needed for optimal epithelial cell invasion, in analogy with observations made for SPI-1^OFF^ and SPI-1^ON^ cells [3].

### 3.3. Growth of Salmonella Cells with Different Patterns of SPI-1 and Flagellar Expression

Expression of *Salmonella* pathogenicity island 1 (SPI-1) is known to retard growth, thus imposing a "fitness cost" associated to synthesis of the T3SS invasion effectors [7]. Based on this antecedent, growth of *S.* Typhimurium cells harboring GFP (to monitor SPI-1 expression) and mCherry (to monitor expression of the flagellar network) was monitored by time-lapse microscopy (Figure 3 and Appendix A). In the experiment summarized in Figure 3, a SPI-1^OFF^ Flag^OFF^ cell formed a colony after 5 h (black arrow) while SPI-1^ON^ Flag^ON^ individual cells (white arrows) did not form colonies. In turn, SPI-1^OFF^ Flag^ON^ and SPI-1^ON^ Flag^OFF^ cells (white circles) only began to divide when either the SPI-1 or the flagellar regulon had been switched to OFF, thus yielding SPI-1^OFF^ Flag^OFF^ cells. Therefore, while the SPI-1^ON^ state causes growth retardation as previously described [7], simultaneous Flag^ON^ and SPI-1^ON^ states cause growth arrest. The SPI-1^ON^ and Flag^ON^ phenotypes do not disappear immediately when cells resume growth, and switching to OFF is observed after a few rounds of division (see Appendix A). 

## 4. Discussion

Coordinated expression of multiple cellular systems is essential for *Salmonella* Typhimurium virulence. At the stage of epithelial cell invasion in the intestine, expression of pathogenicity island 1 (SPI-1) and of the flagellar gene network is regulated by interacting transcription factors [4,5,18,21,22]. It is also well known that both the flagellar regulon and SPI-1 exhibit bistable expression [1,2,3,7,8,15,16,17,18,23,24,25]. None of these previous studies analyzed simultaneous expression of both systems in single cells, which was the main goal of this study. Indeed, we have used fluorescent fusions to analyze the expression of SPI-1 and the flagellar network by microscopy (including time lapse), flow cytometry and cell sorting. Use of cell sorting circumvents the problems found in tissue culture studies using mutant strains, due to the fact that mutations that prevent biosynthesis of flagella often have pleiotropic phenotypes [26].

Simultaneous monitoring of OFF and ON states in SPI-1 and in the flagellar regulon has permitted the visualization of four subpopulations (SPI-1^OFF^ Flag^OFF^, SPI-1^OFF^ Flag^ON^, SPI-1^ON^ Flag^OFF^, and SPI-1^ON^ Flag^ON^), indicating that independent switching occurs (Figure 1). Hence, under gut-like growth conditions, *Salmonella* populations appear to contain four distinct types of cells. 

An unsuspected consequence of the formation of four bacterial lineages was observed when sorted subpopulations were tested in invasion assays in vitro: none of the four subpopulations detected by flow cytometry was highly invasive (Figure 2). This observation raises the possibility that Flag^OFF^ cells may contribute to optimal invasion, in a fashion perhaps analogous to the involvement of SPI-1^OFF^ cells [3,27]. A tentative explanation, speculative at this stage, is that evolution might have endowed Flag^OFF^ cells with hitherto unknown traits that favor invasion. The existence of such traits might explain why the intriguing contribution of Flag^OFF^ cells to invasion is observed in cultured epithelial cells, that is, in the absence of immune responses. In fact, an advantage of the reductionist approach of this study may be the ability to observe interactions between flagella and epithelial cells in the absence of inflammation-related events triggered by the presence of flagella [13,14,28].

Expression of the flagellar regulon enhances the growth defect of SPI-1^ON^ cells; in fact, simultaneous expression of SPI-1 and the flagellar network causes growth arrest (Figure 3). One may thus understand why the subpopulation of SPI-1^ON^ Flag^ON^ cells is relatively small (Figure 1). In contrast, the SPI-1^OFF^ Flag^ON^ subpopulation is large. Because SPI-1^OFF^
*Salmonellae* can invade epithelial cells [3,27], possession of flagella might enhance the ability of such cells to find appropriate invasion sites.

In summary, single cell analysis adds novel features to the known roles of SPI-1 and the flagellar regulon in *Salmonella* invasion. The main observation is that double bistability occurs, producing four subpopulations. The possibility that SPI-1^OFF^ Flag^OFF^, SPI-1^OFF^ Flag^ON^, SPI-1^OFF^ Flag^ON^, and SPI-1^ON^ Flag^ON^ subpopulations cells differ in additional traits besides the presence or absence of the SPI-1 T3SS and flagella may be supported by the pleiotropic control of gene expression exerted by the master flagellar regulator, FlhDC [29]. In analogy with other bistable systems [30,31,32], phenotypic diversification may be advantageous over a deterministic, uniform response by bringing up division of labor and/or other benefits.

## 5. Conclusions

SPI-1 and the flagellar regulon are bistable systems with independent switching, thereby producing four subpopulations: SPI-1^OFF^ Flag^OFF^, SPI-1^OFF^ Flag^ON^, SPI-1^ON^ Flag^OFF^ and SPI-1^ON^ Flag^ON^. However, none of the four subpopulations is highly invasive, suggesting that Flag^OFF^ cells might contribute to optimal invasion as previously described for SPI-1^OFF^ cells. Expression of the flagellar regulon enhances the growth impairment previously described in SPI-1^ON^ cells, and growth resumption in SPI-1^ON^ Flag^ON^ cells requires switching to both SPI-1^OFF^ and Flag^OFF^ states. 

## Figures and Tables

**Figure 1 microorganisms-09-00210-f001:**
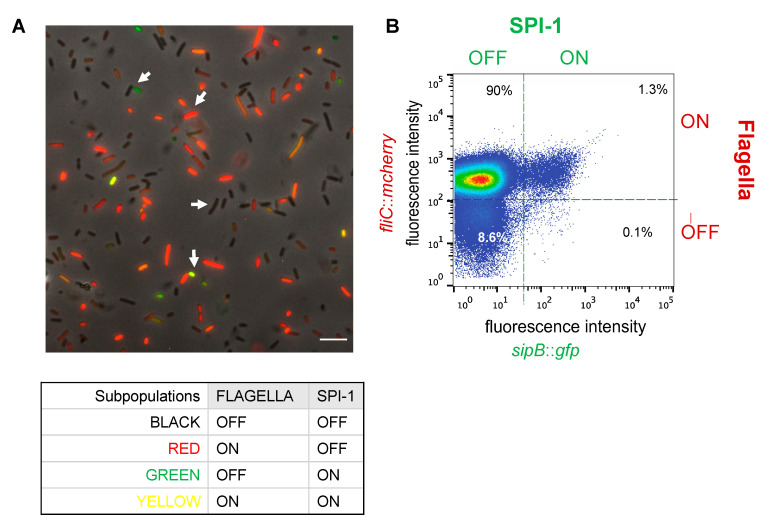
Single cell analysis of expression of SPI-1 (invasion) and flagellar (motility) genes. (**A**) A representative image of *S.* Typhimurium strain SV8533 (*sipB*::*gfp fliC*::*mCherry)* grown at 37 °C in lysogeny broth (LB) under invasive conditions, visualized by fluorescence microscopy with a 100× objective. SPI-1 and flagellar genes are tagged with GFP and mCherry, respectively. Examples of Table 5 μm. (**B**) GFP and mCherry fluorescence intensity distribution in an LB culture of strain SV8533 under invasive conditions, monitored by flow cytometry.

**Figure 2 microorganisms-09-00210-f002:**
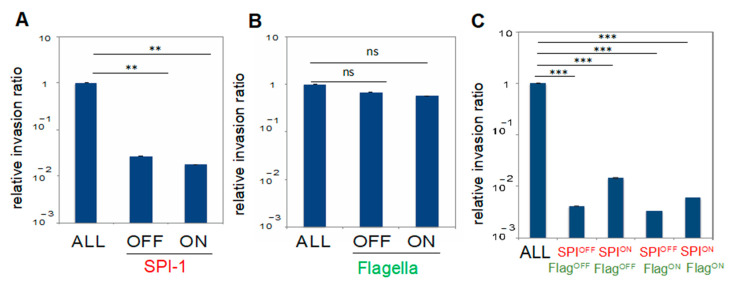
Invasion of epithelial cells by subpopulations expressing invasion and motility determinants (strain SV8533, *sipB*::*gfp* and *fliC*::*mCherry*). (**A**) Invasion rates of sorted SPI-1^OFF^ and SPI-1^ON^ subpopulations. Assays were performed using aliquots from the whole bacterial population after passage through the cell sorter (“ALL”) and from SPI-1^ON^ and SPI-1^OFF^ subpopulations. (**B**) Invasion rates of sorted Flag^OFF^ and Flag^ON^ subpopulations. Culture aliquots used in the assays were as above. (**C**) Invasion rates of sorted SPI-1^OFF^ Flag^OFF^, SPI-1^OFF^ Flag^ON^, SPI-1^OFF^ Flag^ON^, and SPI-1^ON^ Flag^ON^ subpopulations. An aliquot from the whole bacterial population after passage through the cell sorter (“ALL”) was included in the invasion assays. In all panels, averages and standard deviations from >3 independent experiments are shown. Statistical indications: ns, not significantly different; *** significantly different, *p* < 0.001; ** significantly different, *p* < 0.01.

**Figure 3 microorganisms-09-00210-f003:**
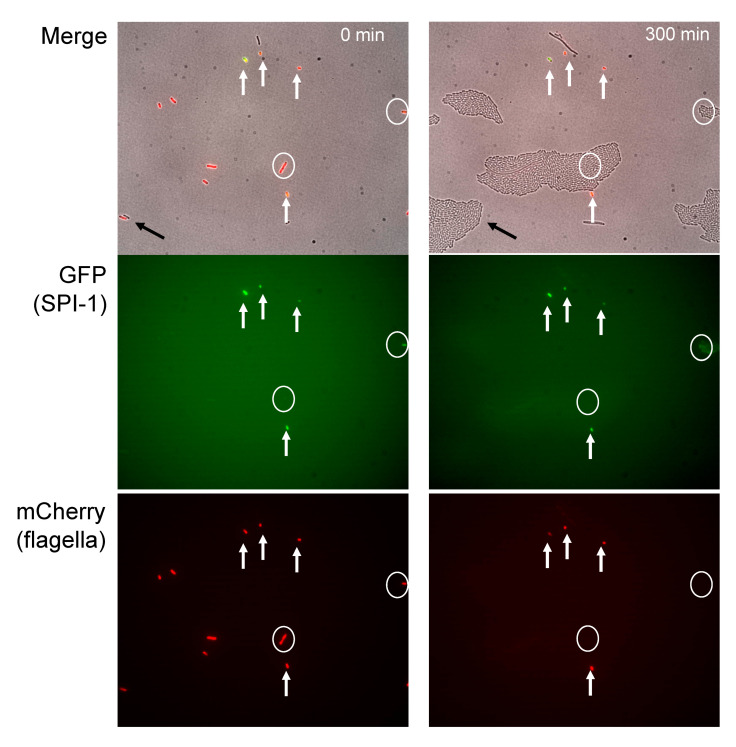
Time lapse microscopy analysis of growth of *S.* Typhimurium cells that differ in the expression of invasion and flagellar genes. Image from a typical time-lapse microscopy experiment with strain SV8533 (*sipB*::*gfp* and *fliC*::*mCherry*) (left panel) and after growth on an agar pad at 37 °C Figure 300 min (right panel). Bacteria were imaged to detect SPI-1 (*sipB* gene) and flagellar (*fliC* gene) expression (green and red fluorescence, respectively) and growth (phase contrast, 1 frame/10 min). Flag^OFF^ SPI-1^OFF^ individual cells are represented as a black arrow, Flag^ON^ SPI-1^ON^ individual cells as white arrows, and Flag^ON^ SPI-1^OFF^ and Flag^OFF^ SPI-1^ON^ cells as white circles.

## Data Availability

Not applicable.

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
