# Peer review of "Single Cell Analysis of Bistable Expression of Pathogenicity Island 1 and the Flagellar Regulon in Salmonella enterica"

_microorganisms, 2021, doi:10.3390/microorganisms9020210_

Round 1

Reviewer 1 Report

Maria and colleagues study the invasion of different subpopulations of Salmonella Typhimurium by using FACS sorter and time-coursed fluorescent microscope. The authors fuse GFP to the SipB, a translocase of Type III secretion system to monitor the SPI-1 expression on/off state. Similarly, the gene of mcherry was inserted into the fliC, encoding flagellar flagellin gene to monitor flagellar on/off state. The authors established a well-suitable system for monitoring on/off state in SPI-1 and flagellar system. The main concern is that how could you claim that Flag OFF cells might contribute to optimal invasion. As Figures 2B shows, no significant difference is visible between the Flag-on and Flag-off groups. Similarly, Flag-off does not clearly improve the invasion ratio in Figures 2C. A few comments are listed below: 

1, Line 13, SPI-1ON Flag OFF. The same typo is at several different lines. 

2, Line 58, microbiologists in the field always use S. Typhimurium as the short name. 

3, Line 98~101: for describing gene, gfp; for describing protein, GFP. 

4, Line 109, Strain SV8533 (sipB::gfp fliC::mCherry)  

5, Line 113, objective lens, 

6, Line 123, with GFP.  

7, Line 121, In the fluorescent picture, there are few cells shown in orange color. Thus, several cells were not in the focus plane. To unambiguously describe 4 subpopulations from the fluorescent image, I recommend you take a Z-stack and further process it into a 2D projection to represent 4 subpopulations. 

8, Line 156, The figure legend should be like : (A) a representative fluorescent image containing 4 subpopulations……; Scale bar is required; Arrow stands for……; Fig. 1B fliC::mcherryand sipB::gfp in Italic 

9, Fig. 3 the S.tm circled in the right side showing both red and green color stands for both SPI-1 and Fla are at ON state. Furthermore, this subpopulation has formed a colony and thus, would not support your claim that Fla and SPI-1 being at the ON state cause the arrest of cell growth. 

Reviewer 2 Report

In this manuscript, Sanchez-Romero and Casadesus describe the development and analysis of Salmonella strains with all possible combinations of switched (on/off) pathogenicity island (SPI-1) and flagella (FLAG) genes. Their findings suggest that FLAG-Off cells may contribute to cellular invasion by Salmonella in ways that are similar to what others have previously described for SPI-1-Off cells. This is novel and somewhat surprising finding. The data is presented in a generally clear and concise manner. Minor comments include the following:

In the single-cell analysis (section 3.1), it would be helpful to show the percentage of each gated cell type in the cell sorting plots. Also, it would be interesting to know how these proportions compare between the "invasive" conditions used by the authors and normal laboratory conditions.

In section 3.2, it is difficult to follow the text without specific guidance on which panel in Figure 2 to focus on. 

In the Discussion, “This observation raises the possibility that FlagOFF cells may contribute to optimal invasion, in a fashion perhaps analogous to the involvement of SPI-1OFF cells” – it was mentioned in the introduction that the mixture of SPI-1 states is important for the invasion but nowhere it was mentioned what is a working model, what could be the reason for better results when we have mixture of states compared to only SPI-1 ON. That could than make clearer what is that “analogous fashion”.

In the Abstract, SPI-1-Off/Flag-On is mentioned twice; SPI-1-On/Flag-Off is missing.
